# BOUNDARY EFFECTS IN CNNS: FEATURE OR BUG?

## ABSTRACT

Recent studies have shown that the addition of *zero* padding drives convolutional neural networks (CNNs) to encode a significant amount of absolute position information in their internal representations, while a lack of padding precludes position encoding. Additionally, various studies have used image patches on background canvases (e.g., to accommodate that inputs to CNNs must be rectangular) without consideration that different backgrounds may contain varying levels of position information according to their color. These studies give rise to deeper questions about the role of boundary information in CNNs, that are explored in this paper: (i) What boundary heuristics (e.g., padding type, canvas color) enable optimal encoding of absolute position information for a particular downstream task?; (ii) Where in the latent representations do *boundary effects* destroy semantic and location information?; (iii) Does encoding position information affect the learning of semantic representations?; (iv) Does encoding position information always improve performance? To provide answers to these questions, we perform the largest case study to date on the role that padding and border heuristics play in CNNs. We first show that zero padding injects optimal position information into CNNs relative to other common padding types. We then design a series of novel tasks which allow us to accurately quantify *boundary effects* as a function of the distance to the border. A number of semantic objectives reveal the destructive effect of dealing with the border on semantic representations. Further, we demonstrate that the encoding of position information improves separability of learned semantic features. Finally, we demonstrate the implications of these findings on a number of real-world tasks to show that position information can act as a feature or a bug.

## 1 INTRODUCTION

One of the main intuitions behind the success of CNNs for visual tasks such as image classification (Krizhevsky et al., 2012; Simonyan & Zisserman, 2015; Szegedy et al., 2015; Huang et al., 2017), video classification (Karpathy et al., 2014; Yue-Hei Ng et al., 2015; Carreira & Zisserman, 2017), object detection (Ren et al., 2015; Redmon et al., 2016; He et al., 2017), generative image models (Brock et al., 2018), and semantic segmentation (Long et al., 2015; Noh et al., 2015; Chen et al., 2017; 2018), is that convolutions add a visual inductive bias to neural networks that objects can appear anywhere in the image. To accommodate the finite domain of images, manual heuristics (e.g., padding) have been applied to allow the convolutional kernel's support to extend beyond the border of an image and reduce the impact of the *boundary effects* (Wohlberg & Rodriguez, 2017; Tang et al., 2018; Liu et al., 2018a; Innamorati et al., 2019; Liu et al., 2018b). Recent studies (Pérez et al., 2019; Islam et al., 2020; Kayhan & Gemert, 2020) have shown that zero padding allows CNNs to encode absolute position information despite the presence of pooling layers in their architecture (e.g., global average pooling). In our work, we argue that the relationship between *boundary effects* and absolute position information extends beyond zero padding and has major implications in a CNN's ability to encode confident and accurate semantic representations (see Fig. 1).

An unexplored area related to boundary effects is the use of *canvases* (i.e., backgrounds) with image patches (see Fig. 1, top row). When using image patches in a deep learning pipeline involving CNNs, the user is required to paste the patch onto a background due to the constraint that the image must be rectangular. Canvases have been used in a wide variety of domains, such as image generation (Gregor et al., 2015; Huang et al., 2019), data augmentation (DeVries & Taylor, 2017), image inpainting (Demir & Unal, 2018; Yu et al., 2018), and interpretable AI (Geirhos et al., 2018; Esser et al., 2020). To the best of our knowledge, this paper contains the first analysis done on canvas value selection.

In other works, the canvas value is simply chosen based on the authors intuition. Given the pervasiveness of CNNs in a multitude of applications, it is of paramount importance to fully understand what the internal representations are encoding in these networks, as well as isolating the precise reasons that these representations are learned. This comprehension can also allow for the effective design of architectures that overcome recognized shortcomings (e.g., residual connections (He et al., 2016) for the vanishing gradient problem). As boundary effects and position information in CNNs are still largely not fully understood, we aim to provide answers to the following hypotheses which reveal fundamental properties of these phenomenon:

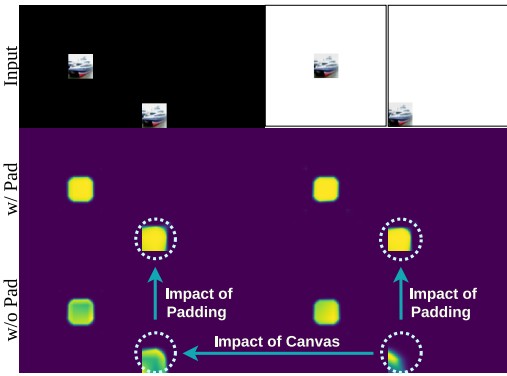

Figure 1: An illustration of how CNNs use position information to resolve boundary effects. We place CIFAR-10 images in random locations on a canvas of 0's (black) or 1's (white). We evaluate if a ResNet-18, trained w/ or w/o padding for semantic segmentation, can segment the image region. Surprisingly, performance is improved when either zero padding or a black canvas is used, implying position information can be exploited from border heuristics to reduce the boundary effect. Colormap is 'viridis'; yellow is high confidence.

**Hypothesis I: Zero Padding Encodes Maximal Absolute Position Information:** Does zero padding encode maximal position information compared to other padding types? We evaluate the amount of position information in networks trained with different padding types and show zero padding injects more position information than common padding types, e.g., reflection, replicate, and circular.

**Hypothesis II: Different Canvas Colors Affect Performance:** Do different background values have an effect on performance? If the padding value at the boundary has a substantial effect on a CNNs performance and position information contained in the network, one should expect that canvas values may also have a similar effect.

**Hypothesis III: Position information is Correlated with Semantic Information:** Does a network's ability to encode absolute position information affect its ability to encode semantic information? If zero padding and certain canvas colors can affect performance on classification tasks due to the increased position information, we expect that the position information is correlated with a networks ability to encode semantic information. We demonstrate that encoding position information improves the robustness and separability of semantic features.

**Hypothesis IV: Boundary Effects Occur at All Image Locations:** Does a CNN trained without padding suffer in performance solely at the border, or at all image regions? How does the performance change across image locations? Our analysis reveals strong evidence that the border effect impacts a CNN's performance at *all regions* in the input, contrasting previous assumptions (Tsotsos et al., 1995; Innamorati et al., 2019) that border effects exist solely at the image border.

**Hypothesis V: Position Encoding Can Act as a Feature or a Bug:** Does absolute position information always correlate with improved performance? A CNN's ability to leverage position information from boundary information could hurt performance when a task requires translation-invariance, e.g., texture recognition; however, it can also be useful if the task relies on position information, e.g., semantic segmentation.

To give answers to these hypotheses (hereon referred to as **H-X**), we design a series of novel tasks as well as use existing techniques to quantify the location information contained in different CNNs with various settings. In particular, we introduce location dependant experiments (see Fig. 2) which use a grid-based strategy to allow for a per-location analysis of absolute position encoding and performance on semantic tasks. The per-location analysis plays a critical role in representing the boundary effects as a function of the distance to the image border. We also estimate the number of dimensions which encode position information in the latent representations of CNNs. Through these experiments we show both quantitative and qualitative evidence that boundary effects have a substantial effect on CNNs in surprising ways and then demonstrate the practical implications of these findings on multiple real-world applications. Code will be made available for all experiments.

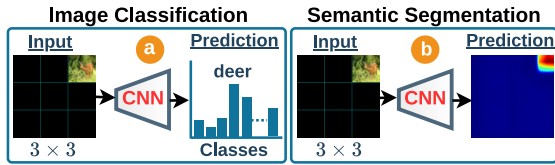

Figure 2: We consider two *location dependant* tasks designed to investigate the boundary effects in CNNs. A random image is placed on a random grid location and the CNN predicts either $C$ class logits (a: classification), or $C$ class logits for each pixel (b: segmentation).

Table 1: Position encoding results w/ metrics SPC↑: high is better and MAE↓: low is better, w/ different padding types. † denotes zero-padding based methods.

| Padding | Horizontal | | Gaussian | |
|---|---|---|---|---|
| | SPC↑ | MAE↓ | SPC↑ | MAE↓ |
| Zero Pad† | .406 | .216 | .591 | .146 |
| Partial† | .424 | .213 | .604 | .144 |
| Circular | .296 | .236 | .455 | .165 |
| Replicate | .218 | .241 | .396 | .173 |
| Reflect | .212 | .242 | .409 | .172 |
| w/o Pad | .204 | .243 | .429 | .168 |

## 2 ABSOLUTE POSITION INFORMATION IN CNNS

**What Type of Padding Injects Optimal Location Information?** With the ultimate goal of revealing characteristics that determine the impact that *boundary effects* plays in CNNs with respect to absolute position information, we first determine which commonly used padding type encodes the maximum amount of absolute position information. We evaluate the ability of different padding types (i.e., zero, circular, reflection, and replicate) to encode absolute position information by extending the experiments from (Islam et al., 2020), which only considered zero padding. We first train a simplified VGG network (Simonyan & Zisserman, 2015) with five layers (VGG-5, see Appendix A.2 for implementation details) on Tiny ImageNet (Le & Yang, 2015) for each padding type. We follow the settings in (Islam et al., 2020): a read-out module, trained using DUT-S (Wang et al., 2017) images, takes the features from a frozen VGG-5 model's last layer, pre-trained on Tiny ImageNet, and predicts a gradient-like position map (see top row in Table. 1). We experiment with two position maps, which are the same for every image: (i) 'horizontal' and (ii) 'Gaussian'. These gradient-like position maps change smoothly from 0 to 1, from the dark-blue to yellow, respectively. For a fair comparison with (Islam et al., 2020), we report results using Spearman Correlation (SPC) and Mean Absolute Error (MAE) with input images from PASCAL-S (Li et al., 2014). From Table 1, it is clear that zero padding delivers the strongest position information, compared with replicate, boundary reflection, and circular padding, supporting **H-I**. Note that partial convolution (Liu et al., 2018a) still pads with zeros, but re-weights the output of the convolution based on how many zeros are padded. Thus, position information is still encoded when partial convolutions are used. Interestingly, circular padding is often the second most capable padding type. We conjecture this is because circular padding takes values from the opposite side of the image where the pixel values are typically less correlated than the directly neighbouring pixels. Thus, circular padding often has a value transition at the border, contrasting reflection and replicate which offer little or no signal to the CNN regarding the whereabouts of the image border.

## 3 LOCATION DEPENDANT TASKS FOR POSITIONAL ANALYSIS

We begin by describing our experimental settings and the implementation details for the proposed location dependant experiments with grid-based inputs. These experiments are used to analyze the border effects with respect to position information encoded in CNNs. These consist of location dependant image classification (Fig. 2 (a) and Sec. 3.1), and segmentation (Fig. 2 (b) and Sec. 3.2), under different canvas color settings. Our experiments are designed with the goal of determining, for different canvas colors (**H-II**), where in the input CNNs suffer from the border effect (**H-IV**), and how the encoding of position information affects the learning of semantic features (**H-III**).

**Experimental Settings and Implementation Details.** Our image classification and segmentation experiments use 'location dependant' inputs (see Fig. 2 above and Fig. 9 in the appendix for more detailed examples). The input is a colored canvas (the colors used are *Black* $[0, 0, 0]$, *White* $[1, 1, 1]$, and the CIFAR-10 dataset (Krizhevsky et al., 2014) *Mean* $[0.491, 0.482, 0.446]$) with an image patch randomly placed on a $k \times k$ grid. Unless mentioned otherwise, we use CIFAR-10 for all experiments. Given a $32 \times 32$ CIFAR-10 training image as the image patch, we *randomly* choose a grid location, $L$, and place the CIFAR-10 training sample in that location. For example, in the case of a $k \times k$ grid, the size of the grid canvas is $32k \times 32k$, where each grid location has a size of $32 \times 32$ and $k^2$ total locations (see Fig. 9 in the appendix). All experiments are run for $k \in \{3, 5, 7, 9, 11, 13, 15\}$. To ensure a fair comparison between grid locations, the evaluation protocol consists of running the entire validation set of CIFAR-10 on *each individual* grid location (i.e., we run the validation set $k^2$ times

Table 2: Location dependant (a) **image classification** and (b) **semantic segmentation** accuracy on CIFAR-10 under zero/no padding settings and canvas colors **B**lack, **W**hite, and **M**ean.

| | Padding | Image Classification | | | | | | | Image Segmentation | | | | | | |
|---|---|---|---|---|---|---|---|---|---|---|---|---|---|---|---|
| | | 3×3 | 5×5 | 7×7 | 9×9 | 11×11 | 13×13 | 15×15 | 3×3 | 5×5 | 7×7 | 9×9 | 11×11 | 13×13 | 15×15 |
| **B** | **Zero Pad** | 78.4 | 74.9 | 74.3 | 76.2 | 76.9 | 75.2 | 76.9 | 68.2 | 67.8 | 67.8 | 67.7 | 64.0 | 63.4 | 60.3 |
| | w/o Pad | 76.4 | 74.0 | 67.6 | 67.5 | 67.4 | 60.3 | 47.2 | 59.5 | 53.5 | 51.4 | 50.4 | 48.1 | 48.3 | 45.6 |
| **W** | **Zero Pad** | 81.2 | 80.4 | 80.3 | 80.3 | 80.3 | 80.1 | 80.0 | 69.0 | 64.3 | 63.2 | 54.8 | 49.1 | 45.9 | 47.4 |
| | w/o Pad | 79.8 | 76.3 | 41.6 | 34.2 | 36.6 | 24.9 | 19.0 | 57.7 | 32.8 | 31.5 | 26.4 | 23.1 | 17.2 | 19.2 |
| **M** | **Zero Pad** | 81.3 | 80.5 | 78.9 | 78.9 | 78.8 | 80.4 | 80.2 | 66.1 | 63.8 | 57.5 | 55.0 | 51.6 | 49.5 | 42.5 |
| | w/o Pad | 81.1 | 77.5 | 68.8 | 34.8 | 31.6 | 32.8 | 26.2 | 42.0 | 35.4 | 29.6 | 26.3 | 24.1 | 23.9 | 24.5 |

for a single validation epoch). We then average the performance over all grid locations to obtain the overall accuracy. We report classification and segmentation accuracy in terms of precision and mean intersection over union (mIoU), respectively. We use a ResNet-18 network trained from scratch, unless stated otherwise. ResNets with no padding are achieved by setting the padding size to zero in the convolution operation. For fair comparison between the padding and no padding baseline, we use bilinear interpolation (see Appendix A.1 for discussion) to match spatial resolutions between the residual output and the feature map for the no padding case, which was not accounted for in previous work (Kayhan & Gemert, 2020).

## 3.1 LOCATION DEPENDANT IMAGE CLASSIFICATION

We investigate whether CNNs trained with and w/o padding are equally capable of exploiting absolute position information to predict the class label in all image locations, with respect to the distance from the image boundary and for variable grid sizes. The location dependant image classification experiment is a multi-class classification problem, where each input has a single class label and the CNN is trained using the multi-class cross entropy loss (see Fig. 2 (a)). Therefore, the network must learn semantic features *invariant to the patch location*, to reach a correct categorical assignment.

Table 7 (left) shows the location dependant image classification results. For all canvases, the networks trained with padding are more robust to changes in grid sizes. In contrast, models trained w/o padding significantly drop in performance with the increase of grid size, as position information is lost and boundary information cannot be exploited. Further, the canvas colors seem to have a noticeable effect on classification performance (**H-II**) as the white and mean canvases have a much more significant performance drop than the black as the grid size increases. The difficulty in separating image semantics from the background signal is due to non-zero canvases creating noisy activations at regions near the image patch border, which is explored further in Section 5.

## 3.2 LOCATION DEPENDANT IMAGE SEGMENTATION

The experiment in this section examines similar properties as the previous location dependant image classification, but for a dense labelling scenario. This task is simply a multi-class per-pixel classification problem, where each pixel is assigned a single class label. We follow the same grid strategy as classification to generate a training sample. Since CIFAR-10 is a classification dataset and does not provide segmentation ground-truth, we generate synthetic ground-truth for each sample by assigning the class label to all the pixels in the grid location where the image belongs to (see Fig. 2 (b)). Following existing work (Chen et al., 2017), we use a per-pixel cross entropy loss to train the network. For evaluation, we compute mIoU at per grid location and take the average to report results.

Image segmentation results are shown in Table 7 (right). A similar pattern is seen as the classification experiment (Sec. 3.1). Networks trained with padding consistently outperform networks trained w/o padding, and the difference grows larger as the grid size increases. Contrasting the classification experiment, the performance of networks with padding decreases slightly as the grid size increases. The reason for this is that the mIoU metric is averaged across all categories including the background, so object pixels are equally weighted in the mIoU calculation even though the ratio of background pixels to object pixels increases dramatically for larger grid sizes. For the no padding case, we observe similar patterns to the classification experiment as the white and mean canvas scenarios suffer more from a large grid size than the black canvas case. This finding further suggests that, independent of the task, a black canvas injects more location information to a CNN (**H-II**), regardless of the semantic difficulty, than a white or mean colored canvas, which is further explored in Sec. 5.

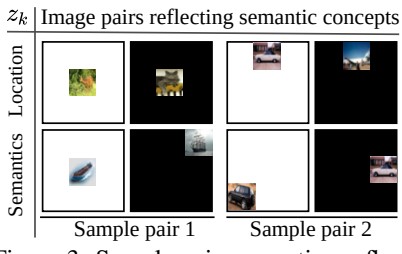

Figure 3: Sample pair generation reflecting two semantic concepts.

Table 3: Dimensionality estimation (%) of two semantic concepts (location and semantic category) under different tasks and settings.

| Canvas | Grid | Padding | Segmentation | | Classification | |
|---|---|---|---|---|---|---|
| | | | $z_{\text{Location}}$ | $z_{\text{Class}}$ | $z_{\text{Location}}$ | $z_{\text{Class}}$ |
| Black | $7 \times 7$ | **Zero Pad** | **15.2%** | **14.9%** | **12.7%** | **12.6%** |
| | | No Pad | 12.7% | 12.8% | 12.1% | 11.9% |
| White | $7 \times 7$ | **Zero Pad** | **12.5%** | **12.3%** | **12.2%** | **12.1%** |
| | | No Pad | 10.9% | 10.9% | 11.5% | 11.6% |

## 4 INTERPRETING REPRESENTATIONS FOR DIMENSIONALITY ESTIMATION

Previous works (Bau et al., 2017; Esser et al., 2020) proposed various mechanisms to interpret different semantic concepts from latent representations by means of quantifying the number of neurons which encode a particular semantic factor, $k$. Given a pretrained CNN encoder $\mathbb{E}(I) = z$ where $z$ is a latent representation and given an image pair $(I^a, I^b) \sim p(I^a, I^b | k)$ which are similar in the $k$-th semantic concept, we aim to estimate the dimensionality of the semantic factor, $z_k$, that represents this concept in the latent representation. A positive mutual information between $I^a$ and $I^b$ implies a similarity of $I^a$ and $I^b$ in the $k$-th semantic concept, which will be preserved in the latent representations $\mathbb{E}(I^a)$ and $\mathbb{E}(I^b)$, only if $\mathbb{E}$ encodes the $k$-th semantic concept. Following (Esser et al., 2020), we approximate the mutual information between $\mathbb{E}(I^a)$ and $\mathbb{E}(I^b)$ with the correlation of each dimension in the latent representation (see Sec. A.5 in Appendix for details).

We generate image pairs which share one of two semantic concepts: (i) *location* or (ii) *semantic class*. For example, the image pair sharing the location factor (see Fig.3 top row) differs in the class and canvas color, while the pair on the bottom row shares the semantic class but differs in canvas color and location. With this simple generation strategy, we can accurately estimate the number of dimensions in the latent representation which encodes the $k$-th semantic factor. Note that the remaining dimensions not captured in either the location or semantic class is allocated to the *residual* semantic factor, which by definition will capture all other variability in the latent representation, $z$.

Table 3 shows the estimated dimensionality for the semantic factors *location* and *class*. The latent representation used is the last stage output of a ResNet-18 before the global average pooling layer. We used the networks from Sec. 3 which are trained for segmentation (left) and classification (right) with the appropriate background (i.e., black on the top and white on the bottom row) and grid settings. The results clearly show that networks trained with zero-padding contain more dimensions which encode the semantic factor 'location' (**H-I**). Further, Table 3 shows that there is a *positive correlation* between the encoding of location and the encoding of semantics, i.e., a larger number of dimensions encoding location implies a larger number of neurons encoding semantics, supporting **H-III**. More dimensionality estimation results can be found in Sec. A.5.1 in the appendix.

## 5 PER-LOCATION ANALYSIS

In this section, we take advantage of the grid-based learning paradigm and conduct further evaluations on a per-location basis to test **H-I**, **H-II**, **H-III**, and **H-IV**. In particular, we analyze the *relationship* between zero padding and the border effect. We then show quantitative and qualitative results which reveal strong evidence that *zeros*, whether as a canvas or padding, inject maximal location bias.

**Distance-to-Border Analysis: What Input Regions Suffer Most from Border Effects?** First, we analyze the image classification and segmentation results reported in Secs. 3.1 and 3.2, with respect to the distance from the closest border which will allow us to answer this question. To obtain the accuracy at each distance, we average the accuracies over all grid locations with the same distance to the nearest border (e.g., a distance to a border of zero refers to the average accuracy of the outer-most ring of grid locations). Figure 4 (left) shows the accuracy difference between the padding baseline (the blue horizontal line) and the no padding cases. Interestingly, the accuracy difference is higher at grid locations close to the border and decreases towards the image center. This analysis strongly suggests that zero padding significantly impacts the border effect, and injects position information to the network as a function of the object location relative to the distance of the nearest border. In

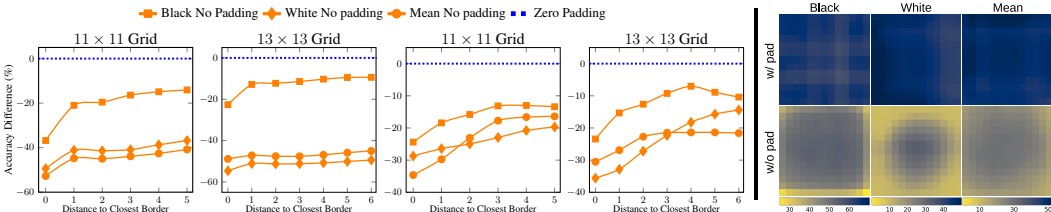

Figure 4: **Left:** Location dependant image classification (left two) and segmentation (right two). Results show the accuracy difference between padding and no padding, at various distances to the border and canvases. **Right**: Heatmap visualization of $13 \times 13$ grid per-location segmentation mIoU on CIFAR-10 (Krizhevsky et al., 2014) under *black* (left), *white* (middle), and *mean* (right) background settings. As can be seen, CNNs without padding have difficulties near the border.

contrast, the no padding case fails to deliver any position information at the border locations which leads to a significant performance drop. Also note that there is a substantial difference in performance at the center of the image, at the farthest distance from the border, supporting **H-IV**. We also visualize the accuracy at all grid locations as a heatmap in Fig. 4 (right) which shows consistency with the results in Fig. 4 (left). As shown in Fig. 4 (right), the networks trained w/ zero padding perform consistently across all grid locations while the networks trained w/o padding perform poorly near the border. Note that of the three canvases for the no padding case, the black canvas yields the lowest drop in relative performance when comparing the center region to locations near the border (**H-II**).

**Are Border Effects Only at the Border?** While intuition might suggest the border effect occurs solely at the border, it is natural to analyze if other regions in the input space also suffer from the border effect. Figure 5 (left) compares filter activations with and without zero padding. Note that filter activations are randomly sampled from the feature map for the specific layer. Activations found near the border propagate less information through the network during the forward pass due to the limited amount of connectivity to downstream layers compared to activations at the center, as discussed in (Tsotsos et al., 1995). Further, the convolution cannot fully overlap the border regions without padding and thus will not recognize objects as well. This phenomenon can be seen in Fig. 5 (bottom-left), where the activations for grid location 5 are significantly reduced in the no padding case. Interestingly, for grid location 13 (i.e., center), there is also a visible difference in the activation space. Here, activations found for the no padding case are blurred and noisy which contrasts the tight square shaped activations when zero padding is used. While border effects mainly impact regions near the border, these results show clear evidence that input locations at the center of the image are also impacted with a lack of padding which is evidence supporting **H-IV**. This also explains the large performance drop at the center of the grid in Fig. 4 (left and right).

**Does Encoding Location Enable the Learning of Semantics?** In Sec. 4, we provided quantitative evidence that reveals the correlation between the number of neurons encoding position and semantic information (**H-III**). We further investigate this phenomenon to see how position information by means of zero padding allows for richer semantics to be learned for the tasks of image classification and semantic segmentation. Figure 5 (right) heatmaps show segmentation predictions for different grid locations, $L$, of a $7 \times 7$ grid. When no padding is used CNNs have difficulty segmenting images

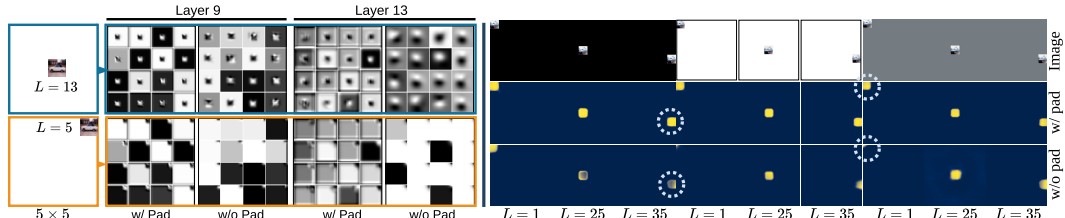

Figure 5: **Left:** Filter activation visualization for the classification task on CIFAR-10 with a *white* background and $5 \times 5$ grid size. It is clear that zero padding provides richer information and larger activations downstream, particularly at locations near the boundary (e.g., $L = 5$). **Right**: Sample predictions of semantic segmentation on different locations of a $7 \times 7$ grid under three background settings. Confidence maps are plotted with the 'cividis' colormap, where yellow and dark blue indicates higher and lower confidence, respectively.

Figure 6: Comparison of **filter activations** for the location dependant segmentation task trained without padding, $5 \times 5$ grid size, $L = 13$, and three canvas colors, black, white, and mean. Notice the large activations in the background region for black, contrasting that of white and mean.

near the border (highlighted with circles in Fig. 5) except when a black canvas is used. However, for locations near the center of the image, reduced position information due to no padding greatly reduces the network's confidence in semantic encodings. In contrast, zero padding is consistent and confident in segmenting objects across all the grid locations and canvas colors. Further, we use t-SNE (Maaten & Hinton, 2008) to visualize the classification logits in Fig. 12 of appendix. The separability of the semantic classes is significantly improved when padding is used, and the effect is particularly pronounced at locations near the border ($L = 7$). This further supports **H-III** that absolute position information, by means of zero padding, enables CNNs to learn more robust semantic features, which in turn allows for greater separability in the prediction logits.

**Canvas Analysis: Why Do Explicit Zeros Inject Location Information?** We now explore what enables CNNs to encode positional information when zeros exist at the boundary (i.e., as padding or canvas (**H-II**)) by analyzing the activations of a network trained for the location dependant segmentation task. For a $k \times k$ grid, the ratio of canvas pixels to total pixels is $\frac{k^2-1}{k^2}$. This implies that the vast majority of labels will be the background class, and therefore the majority of filters should focus on correctly labelling the canvas. To determine if this is true for all canvases, we visualize randomly sampled filter activations (see Fig. 6) for networks trained *without* padding for the location dependant segmentation task. The activations are visualized using the 'gray' colormap, where light and dark intensities denote high and low activations, respectively. Note that the activations are taken from the output of the convolutional layer and are normalized to between $[0, 1]$ before plotting. Even at the earlier layers (e.g., layer 7), there is a clear difference in the patterns of activations. The majority of filters have low activations for the image region, but *high activations for the background region*. In contrast, the *white and mean canvases have mostly low activations for the canvas* but high activations for the image. Interestingly, particularly at layer 17 (the last convolution layer), the activations for the black background are reminiscent of oriented filters (e.g., Gaussian derivative filters) in a number of *different orientations and locations*, indicating they can capture more diverse input signals compared to the white and mean canvases, which consistently activate over the *center* of the input region. Figure 6 clearly demonstrates that zeros at the boundary, in the form of a black canvas, allows easier learning of semantics and absolute position for CNNs compared to other values supporting **H-II**.

# 6 Applicability to Semantic Segmentation, Texture Recognition, Data Augmentation, and Adversarial Robustness

Given the intriguing findings above, it is natural to ask how much the demonstrated phenomenon affects real world tasks with SOTA architectures. More specifically, does encoding position always improve performance or does it cause unwanted effects on certain tasks (**H-V**)?

**Semantic Segmentation.** We now measure the impact of zero padding to segment objects near the image boundary with a strong semantic segmentation network on an automotive-centric dataset. We use the DeepLabv3 (Chen et al., 2017) network and the Cityscapes (Cordts et al., 2016) dataset, trained with different padding types. From Table 4, it is clear that DeepLabv3 with zero padding achieves superior results compared to the model trained without padding or with reflect padding. Additionally, we perform an analysis by computing the mIoU for rectangular ring-like regions (see Fig. 7 (top-left)), between $X\%$ and $Y\%$, where $X$ and $Y$ are relative distances from the border (e.g., $0\% - 5\%$ is the outer most region of the image, while $5\% - 10\%$ is the neighbouring inner $5\%$ region) to quantify the performance decrease from the boundary effect and lack of positional information. From Table 4, the performance drop between the total mIoU (100%) and the border region (0-5%) is more significant for the no padding case and reflect padding case compared to the zero padding case, which agrees with the results found in Sec. 5. This further demonstrates that the absolute position

Table 4: Performance comparison of DeepLabv3 w/ and w/o *padding* for different image regions. Top-left image in Fig. 7 shows outer regions used for this analysis.

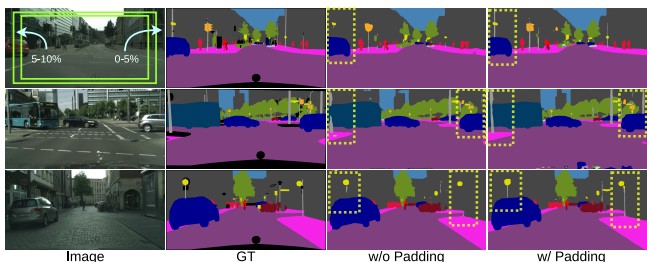

Figure 7: Example predictions on the Cityscapes validation set when training w/ and w/o *padding*. Best viewed zoomed in.

| Method | Evaluation Region mIoU(%) | | | |
|---|---|---|---|---|
| | 0-5% | 5-10% | 10-15% | 100% |
| **Zero Pad** | **72.6** | **72.7** | **73.8** | **74.0** |
| Reflect | 71.9 | 72.0 | 73.7 | 73.9 |
| No Pad | 63.7 | 66.4 | 67.3 | 69.1 |

Table 5: (a) **Texture recognition** results on two datasets with different padding types. (b) **Performance and robustness** of DeepLabv3 variants trained with Cutout (DeVries & Taylor, 2017) using two canvas (**B**lack and **W**hite).

| Padding | GTOS-M | | | DTD | | |
|---|---|---|---|---|---|---|
| | Res34 | Res50 | VGG5 | Res34 | Res50 | VGG5 |
| No Pad | 71.7 | 76.3 | 33.6 | 57.5 | 67.0 | 27.3 |
| Zero Pad | 78.7 | 81.7 | 39.7 | 68.6 | 70.6 | 32.8 |
| **Reflect** | **80.6** | **85.0** | **43.1** | **68.8** | **71.2** | **34.0** |

(a)

| Method | Segmentation | | Robustness | |
|---|---|---|---|---|
| | **B** | **W** | **B** | **W** |
| DLv3-Res50 | 73.9 | 74.1 | 53.7 | **55.8** |
| DLv3-Res101 | 75.5 | 75.2 | 49.8 | **51.9** |

(b)

information due to zero padding improves the performance at all image regions, while reflect padding is not as beneficial at the image boundaries. Figure 7 shows examples of how DeepLabv3 trained with zero padding generates more accurate predictions, particularly near the border of the image. Note that thin or complex objects near the border regions are particularly affected (e.g., light posts). The reason that performance suffers even *with* padding, is the lack of semantic and contextual information near the border, which is not the case for grid-based tasks (Sec. 3) since the image patch contains the entire CIFAR-10 image. Additional results can be found in Sec. A.8 in the Appendix.

**Texture Recognition.** We evaluate three models with three padding types on the task of texture recognition. We use a ResNet-34, ResNet-50, amd VGG-5 trained with zero, reflect, and no padding settings, with the GTOS-Mobile dataset (Xue et al., 2018) and DTD (Cimpoi et al., 2014). We hypothesize that, due to the nature of textures existing in the entire image, position information will not benefit the performance of the CNN. As shown in Table 5 (a), models trained with reflect padding outperform the models trained with zero padding. This result implies that position information is not important for the task of texture recognition. Although no padding has less position information than reflect padding, the CNN suffers from the border effects without padding (see Fig. 5 (left)), which hurts performance significantly (i.e., since the kernel's support does not cover the entire image domain).

**Canvas Analysis: Cutout & Adversarial Robustness.** We investigate the impact of different canvas colors in terms of performance and robustness using a data augmentation strategy, Cutout (DeVries & Taylor, 2017), which simply places a rectangular black mask over random image regions during training. We evaluate DeepLabv3 with two backbones using the Cutout strategy for semantic segmentation on the PASCAL VOC 2012 (Everingham et al., 2010) dataset with black *and* white masks (see Fig. 17 in the appendix for example inputs). We also evaluate the robustness of each model to show which canvas is more resilient to the GD-UAP adversarial attack (Mopuri et al., 2018). Note that the GD-UAP attack is generated based on the image-agnostic DeepLab-ResNet101 backbone. As shown in Table 5 (b), DeepLabv3 trained with white-mask Cutout is significantly more robust to adversarial examples than the black canvas, without sacrificing segmentation performance.

## 7  CONCLUSION

With the goal of answering whether boundary effects are a feature or a bug, we have presented evidence that the heuristics used at the image boundary play a much deeper role in a CNN's ability to perform different tasks than one might assume. By designing a series of location dependant experiments, we have performed a unique exploration into how this connection reveals itself. We

showed that zero padding encodes more position information relative to common padding types (**H-I**) and that zero padding causes more dimensions to encode position information and that this correlates with the number of dimensions that encode semantics (**H-III**). We examined the ability of CNNs to perform semantic tasks as a function of the distance to a border. This revealed the capability of a black canvas to provide rich position information compared to other colors (i.e., White and Mean) (**H-II**). We visualized a number of features in CNNs which showed that boundary effects have an impact on *all* regions of the input (**H-IV**), and highlighted characteristics of border handling techniques which allow for absolute position information to be encoded. This position encoding enables CNNs to learn more separable semantic features which provide more accurate and confident predictions (**H-III**). We conducted these experiments with the following question in mind: Are boundary effects a feature or a bug (**H-V**)? After teasing out the above underlying properties, we were able to validate the hypothesis that different types of padding, levels of position information, and canvas colors, could be beneficial *depending on the task at hand*! To be more clear: the boundary effects can be used to improve performance if you know what to look for, but can also be detrimental to a CNNs performance if not taken into consideration. We strongly believe these findings will allow future researchers to more efficiently propose and implement improved algorithms which deal with boundary effects.

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

# A    APPENDIX

## A.1    NO PADDING RESNET IMPLEMENTATION USING BILINEAR INTERPOLATION

In this work, we attempt to explore the effect of different padding strategies in addition to networks with *no padding*. Our goal is to analyze the most standard settings in computer vision and as ResNets are one of the most common CNN architectures used, we thus aim to remove padding from a ResNet. The main issue with this objective is that the residual block of a ResNet requires padding for the residual connection to match the same feature resolution as the next layer. This question can therefore be reduced to 'how can the input be aligned with the output feature map?'.

To the best of our knowledge, how to match the feature map and the input is an open question. Two simple solutions could be applied: (i) crop the centre region from the input to align with the output, or (ii) interpolate the output feature map to the same size as the input. Before choosing an appropriate operation, we must first consider what the feature map encodes. For dense labelling tasks, the intermediate feature map is believed to retain the spatial relationship because convolutional filters are translation-equivariant.

For instance, most of the one-stage object detectors apply location loss for the bounding box on multiple feature maps, both anchor-based and anchor-free. For example, if a dog is located at $0.1 \times width$ to the right of input, its corresponding region in the feature map will be the same. If this intermediate embedding is translation-equivariant, it is clear that the centre cropping will break this equivariance. If we crop the centre of the input image, the input be aligned with the smaller feature map, the relative position of the dog will be changed. In contrast, interpolation (e.g., bilinear) of the feature map to the same size as the input appears to be a better choice because the whole process should keep a more consistent spatial alignment.

We now validate this choice by conducting a simple toy experiment. We create a black canvas with three channels, (100, 100, 3), the height and width equal 100, (see Fig. 8(a)). Then, we fill four squares with different colours at the four corners where each square has a size of $10 \times 10$. One convolution layer with a kernel size of 21 and a stride of one is applied on the input image with the all weights of the kernel set to one. Note that we only care about the spatial equivariance in this experiment, so the convolutional filter is applied separately on each channel so as to retain the color information.

The output of a padded convolution is shown in Fig. 8(b) ($100 \times 100$), and the output of the convolution w/o padding is shown in Fig. 8(c) ($80 \times 80$). Note that the color is enhanced for visualization purposes, the actual output value will decay gradually towards the center of the feature map, and it will spread out even more when padding is applied. As discussed previously, one can also crop the center of the input to match the feature map (see Fig. 8(e)). However, the cropping operation will excludes the content near the border and the translation-equivariance property no longer holds. Thus, we choose to interpolate the feature map to the input size. As we can see from Fig 8(d), the resized feature map can better match the content of the input.

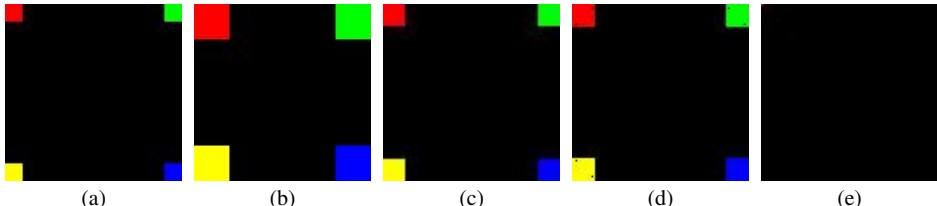

|  (a)  |  (b)  |  (c)  |  (d)  |  (e)  |

Figure 8: (a) the input image, ($100 \times 100$). (b) the output of the convolution with padding, ($100 \times 100$). (c) the output of the convolution w/o padding, ($80 \times 80$). (d) the resized feature map of (c), ($100 \times 100$). (e) the center crop of the input image, ($80 \times 80$).

## A.2    IMPLEMENTATION DEATILS OF VGG-5 NETWORK FOR POSITION INFORMATION

We use a simplified VGG network (VGG-5) for the position encoding experiments in Sec. 2 and texture recognition experiments in Sec. 6. The details of the VGG-5 architecture are shown in Table 6

(in this table we show the VGG-5 network trained on the tiny ImageNet dataset, the VGG-5 network trained on texture recognition has a different input size: $224 \times 224$). Note that the network is trained from scratch. The tiny ImageNet dataset contains 200 classes and each class has 500 images for training and 50 for validation. The size of the input image is $64 \times 64$, a random crop of $56 \times 56$ is used for training and a center crop is applied for validation. The total training epochs is set to 100 with an initial learning rate of $0.01$. The learning rate was decayed at the $60th$ and $80th$ epochs by multiplying the learning rate by a factor of $0.1$. A momentum of $0.9$ and a weight decay of $1e-4$ are applied with the the stochastic gradient descent optimizer. After the pre-training process, a simple read-out module is applied on the pre-trained frozen backbone for position evaluation, following the training protocol as used in Islam et al. (2020). Note that the type of padding strategy is consistent between the pre-training and position evaluation procedures.

Table 6: VGG-5 architecture trained on tiny ImageNet.

| RGB image $x \in \mathbb{R}^{56 \times 56 \times 3}$ |
| :---: |
| Conv2d ($3 \times 3$), Batch Norm, ReLU, MaxPool2d $\rightarrow \mathbb{R}^{28 \times 28 \times 32}$ |
| Conv2d ($3 \times 3$), Batch Norm, ReLU, MaxPool2d $\rightarrow \mathbb{R}^{14 \times 14 \times 64}$ |
| Conv2d ($3 \times 3$), Batch Norm, ReLU, MaxPool2d $\rightarrow \mathbb{R}^{7 \times 7 \times 128}$ |
| Conv2d ($3 \times 3$), Batch Norm, ReLU $\rightarrow \mathbb{R}^{7 \times 7 \times 256}$ |
| Global Average Pooling (GAP) $\rightarrow \mathbb{R}^{1 \times 1 \times 256}$ |
| FC $\rightarrow$ (256, classes) |

### A.3 GRID-BASED INPUTS FOR POSITIONAL ANALYSIS

Figure 9 shows examples of inputs for the location dependant experiments (Sec. 4 in the main paper), and the ground truth for each of the tasks on the right hand side. As previously mentioned, all the experiments were run with three different canvas colors to show the impact of the border effect with regards to backgrounds. For the segmentation ground truth, the ratio of background pixels to object pixels grows exponentially as the grid size increases. However, as the evaluation metric is mean intersection over union (mIoU), the overall performance is averaged between the object classes and the background class, even though the background class makes up the majority of the ground truth labels.

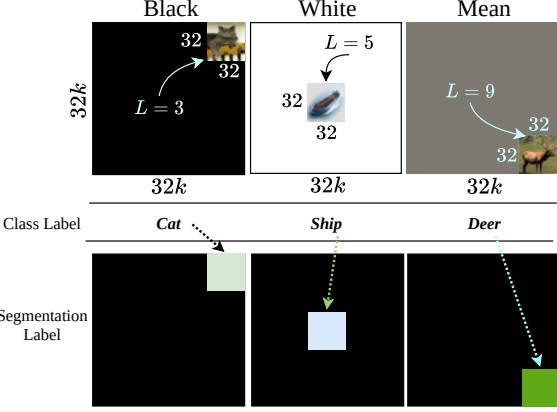

Figure 9: An illustration of our proposed grid settings ($k = 3$) with all three different canvas colors for the location dependant tasks.

## A.4 Location Dependant Image Classification and Segmentation

In this section, we revisit the location dependant image classification and segmentation experiments from Sec. 3 of the main paper, but present them in a visual format (See Fig. 10) to enhance the reader's ability to see patterns in performance as the grid size increases. It is apparent that zero padding provides a massive performance boost for location dependant tasks as larger grid sizes are used. Also, it is apparent that zero padding is particularly helpful as the ratio of background pixels to image pixels increases in the input. For segmentation, the performance decreases even with zero padding; however, a large performance drop is still observed with lack of padding. Additionally, note the increase in performance when a black canvas is used compared with white or mean canvases (see Sec. 5 in the main paper for a discussion on black canvases).

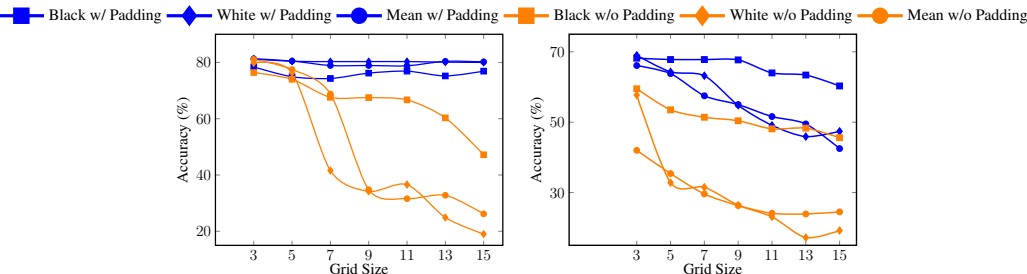

Figure 10: **Location dependant image classification (left) and semantic segmentation (right)**. This figure contains the same results as in Table 2 in the main paper, but visualized as a plot. Note the performance gains when zero padding or a black canvas is used.

**Results using VGG-11 Networks.** We extend the location dependent segmentation experiment by applying a different backbone architecture. We choose VGG-11 network to further validate the correlation between network architectures and boundary effects. Table presents the Image segmentation results. We observe a similar pattern as segmentation experiment (Sec. 3.2). Networks trained with padding consistently outperform networks trained w/o padding, and the difference grows larger as the grid size increases.

Table 7: Location dependant **semantic segmentation** accuracy on CIFAR-10 under zero/no padding and black canvas settings using VGG-11 network.

|   | Padding | Image Segmentation | | | | | | |
|---|---------|------|------|------|------|-------|-------|-------|
|   |         | 3×3  | 5×5  | 7×7  | 9×9  | 11×11 | 13×13 | 15×15 |
| **B** | **Zero Pad** | **66.8** | **66.2** | **64.5** | **63.2** | **56.9** | **56.0** | **52.6** |
|   | w/o Pad | 63.2 | 55.8 | 53.3 | 50.1 | 48.8 | 47.2 | 48.7 |

**Results on reference grid size using ResNet-18 Networks.** We extend the location dependent segmentation experiment by applying different even grid sizes. Table 8 presents the Image segmentation results. We observe a similar pattern as other grid sizes in segmentation experiment (Sec. 3.2).

Table 8: Location dependant **semantic segmentation** accuracy on CIFAR-10 under zero/no padding and black canvas settings using ResNet18 network.

|   | Padding | Segmentation | |
|---|---------|------|------|
|   |         | 4×4  | 6×6  |
| **B** | **Zero Pad** | **68.8** | **68.1** |
|   | w/o Pad | 58.1 | 54.9 |

## A.5 Dimensionality Estimation of Different Semantic Concepts

For an image pair $(I^a, I^b) \sim p(I^a, I^b|k)$ which are similar in the $k$-th semantic concept, following (Esser et al., 2020), we approximate the mutual information with their correlation for each dimension $i$:

$$Correlation_k = C_k = \sum_i \frac{\mathrm{Cov}\left(\mathbb{E}\left(I^a\right)_i, \mathbb{E}\left(I^b\right)_i\right)}{\sqrt{\mathrm{Var}\left(\mathbb{E}\left(I^a\right)_i\right)\mathrm{Var}\left(\mathbb{E}\left(I^b\right)_i\right)}}, \tag{1}$$

We assume that the residual factor has a maximum dimension of $|z|$ (the total dimension of the latent representation) and use the softmax equation to get the resulting dimension:

$$|z_k| = \left\lfloor \frac{\exp C_k}{\sum_{f=0}^{F} \exp C_f} N \right\rfloor, \tag{2}$$

where $|z_k|$ is the dimension of the semantic factor $k$, and $F$ is the total number of semantic factors including the residual factor. Note we do not need an estimate of the absolute mutual information for estimating the proportion of location and semantic dimensions. Only the differences between the mutual information for position and semantic class for image pairs are used to quantify the ratio of location and semantic-specific neurons. Therefore, the relative difference is still meaningful and only the absolute numbers might not be.

### A.5.1 Additional Dimensionality Estimation Results

Table 9 presents additional results of dimensionality estimation for different semantic concepts based on the latent representation (see Table 3 in the main paper). We observe consistent results as zero padding encodes more location and semantic information compared to the no padding case.

Table 9: Dimensionality estimation (%) of two semantic concepts (location and semantic category) under different tasks and settings. Remaining dimensions are assigned to residual factor.

| Canvas | Grid | Padding | Semantic Factors, $z$ | |
|---|---|---|---|---|
| | | | $\|z_{\text{Location}}\|$ | $\|z_{\text{Class}}\|$ |
| Black | 5×5 | Zero Pad | **17.0%** | **16.3%** |
| | | No Pad | 12.6% | 12.8% |
| White | 5×5 | Zero Pad | **16.4%** | **15.4%** |
| | | No Pad | 11.5% | 11.6% |

(a) Semantic Segmentation

| Canvas | Grid | Padding | Semantic Factors, $z$ | |
|---|---|---|---|---|
| | | | $\|z_{\text{Location}}\|$ | $\|z_{\text{Class}}\|$ |
| Black | 9×9 | **Zero Pad** | **11.2%** | **11.2%** |
| | | No Pad | 10.9% | 10.9% |

(a) Image Classification

## A.6 Extended Per-location Analysis

We now present additional 'per-location' results. That is, we take advantage of the location dependant grid-based input and analyze the performance of CNNs at each location on the grid. This is done to reveal the impact of border effects with respect to the absolute location of the object of interest. We first show class-wise performance for the location dependant semantic segmentation task (Sec. A.6.1). Next, we show the performance as a function of the distance to the nearest border by averaging the accuracy over all locations which are a specified number of grid locations away from the nearest border (Sec. A.6.2). We then provide additional t-SNE (Maaten & Hinton, 2008) visualizations to examine the separability of the CNN's learned semantic features (Sec. A.6.3). Finally, we display location dependant semantic segmentation predictions and analyze the impact that border effects have on per-pixel predictions for various input locations (Sec. A.7). Note that all experiments are done with the same settings as Sec. 4 in the main paper, on the CIFAR-10 (Krizhevsky et al., 2014) dataset.

Table 10: **Location dependant image segmentation:** Category-wise mIoU on CIFAR-10 (Krizhevsky et al., 2014) for two different locations under w/ and w/o padding settings and Black and Mean canvas color. The grid size for both canvases is $7 \times 7$.

| Categories | Black | | | | Mean | | | |
|---|---|---|---|---|---|---|---|---|
| | $L = 7$ | | $L = 25$ | | $L = 7$ | | $L = 25$ | |
| | w/ Pad | w/o Pad | w/ Pad | w/o Pad | w/ Pad | w/o Pad | w/ Pad | w/o Pad |
| Background | 0.99 | 0.99 | 0.99 | 0.99 | 0.99 | 0.98 | 0.99 | 0.99 |
| Plane | 0.67 | 0.34 | 0.65 | 0.53 | 0.50 | 0.07 | 0.55 | 0.31 |
| Car | 0.80 | 0.51 | 0.76 | 0.68 | 0.58 | 0.08 | 0.69 | 0.55 |
| Bird | 0.57 | 0.18 | 0.57 | 0.44 | 0.42 | 0.01 | 0.47 | 0.24 |
| Cat | 0.46 | 0.14 | 0.43 | 0.35 | 0.34 | 0.01 | 0.40 | 0.16 |
| Deer | 0.63 | 0.30 | 0.62 | 0.49 | 0.47 | 0.01 | 0.55 | 0.18 |
| Dog | 0.53 | 0.30 | 0.53 | 0.39 | 0.43 | 0.01 | 0.50 | 0.20 |
| Frog | 0.67 | 0.41 | 0.64 | 0.59 | 0.53 | 0.03 | 0.63 | 0.48 |
| Horse | 0.70 | 0.41 | 0.70 | 0.57 | 0.52 | 0.02 | 0.62 | 0.32 |
| Ship | 0.78 | 0.43 | 0.74 | 0.64 | 0.58 | 0.03 | 0.66 | 0.48 |
| Truck | 0.74 | 0.41 | 0.71 | 0.60 | 0.57 | 0.04 | 0.67 | 0.44 |
| **Overall** | **0.66** | **0.40** | **0.67** | **0.57** | **0.54** | **0.12** | **0.61** | **0.40** |

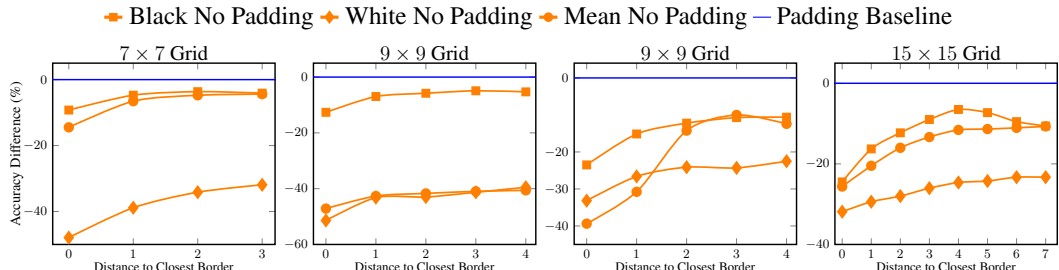

Figure 11: **Location dependant image classification (left two) and segmentation (right two).** Results show the accuracy difference between padding (blue horizontal line) and no padding (orange markers), at various distances to the border and canvas colors.

### A.6.1 PER-LOCATION CATEGORY-WISE MIOU ANALYSIS

Table 10 shows the category-wise mIoU for the location dependant image segmentation task for a $7 \times 7$ grid with black and mean canvases. We show the category-wise performance for a location at the very top right corner ($L = 7$) and at the center of the grid ($L = 25$), which highlights how the encoding of absolute position information affects the learning of semantic representations.

For both locations, the border and the center, zero padding gives a large increase in performance for all classes compared to lack of padding. This is particularly pronounced with a mean canvas, demonstrating how the black canvas explicitly injects position information, even without the use of zero padding. For example, comparing the black and mean canvas at $L = 7$ shows how important absolute position information can be in learning distinct semantic representations. The network trained with a mean canvas has a difficult time learning to segment images at this location when no padding is used and suffers a large drop in performance compared to the black canvas. Some classes even score around $1\%$ mIoU, which implies that the network fails to learn to segment certain classes (i.e., Bird, Cat, Deer, and Dog) with these settings. When zero padding is added (i.e., Mean, w/ padding, $L = 7$), the network achieves a performance boost of between $35\% - 60\%$. When a black canvas is used to inject position information instead (i.e., Black, w/o padding, $L = 7$), the performance gains range from $15\% - 40\%$. Clearly, the encoding of position information, by means of zero padding or a black canvas, has a stark effect on a CNN's ability to learn distinctive semantic

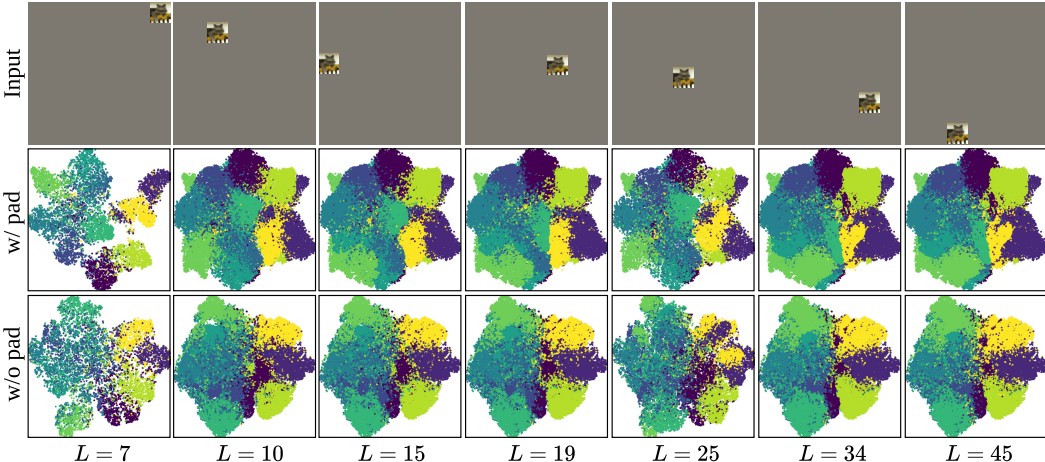

Figure 12: **t-SNE (Maaten & Hinton, 2008) visualization of the CIFAR-10 test set classification logits** for a $7 \times 7$ grid. Examples of a single input are given in the top row, while the embedding visualizes the entire dataset (bottom two rows).

features. We see a similar, but not quite as drastic, pattern at the center of the image, further showing how the boundary effects impact all locations in an image, and not just at the image border.

### A.6.2 DISTANCE TO BORDER PERFORMANCE

Figure 11 shows the performance as a function of the distance to the closest border for all three canvas colors. The networks with zero padding are represented as a blue horizontal line, where the plotted markers show the difference in performance when no padding is used. Consistent with the results in the main paper, locations near the border are on average, much more difficult for networks to classify and segment, particularly as the grid size increases.

### A.6.3 T-SNE VISUALIZATIONS

We use t-SNE (Maaten & Hinton, 2008) to visualize the test set classification logits (from the location dependant classification task) in Fig. 12. Note that the single input examples at the top row are shown merely to highlight the location $L$, and that the second and third rows show embeddings of the entire test set. The separability of the semantic classes is significantly improved when padding is used, and the effect is particularly pronounced at locations near the border ($L = 7$). This further supports the hypothesis that absolute position information, by means of zero padding, enables CNNs to learn more robust semantic features, which in turn allows for greater separability in the predicted logits.

### A.7 LOCATION DEPENDANT IMAGE SEGMENTATION PREDICTIONS

Figure 13 shows predictions of the location dependant image segmentation task for a grid size $k = 5$. We visualize the predictions as a heatmap, where each pixel is colored according to the confidence that the semantic category appears in that pixel's location. We show predictions with padding (left) and without padding (right) for various grid locations, $L$. Note how boundary effects significantly impact locations near the border. In particular, locations in the corners are most affected, as they suffer from boundary effects originating from two borders (e.g., top and left border for $L = 1$). Figure 13 shows predictions of the location dependant image segmentation task for a grid size $k = 5$. We visualize the predictions as a heatmap, where each pixel is colored according to the confidence that the semantic category appears in that pixel's location. We show predictions with padding (left) and without padding (right) for various grid locations, $L$. Note how boundary effects significantly impact locations near the border. In particular, locations in the corners are most affected, as they suffer from boundary effects originating from two borders (e.g., top and left border for $L = 1$).

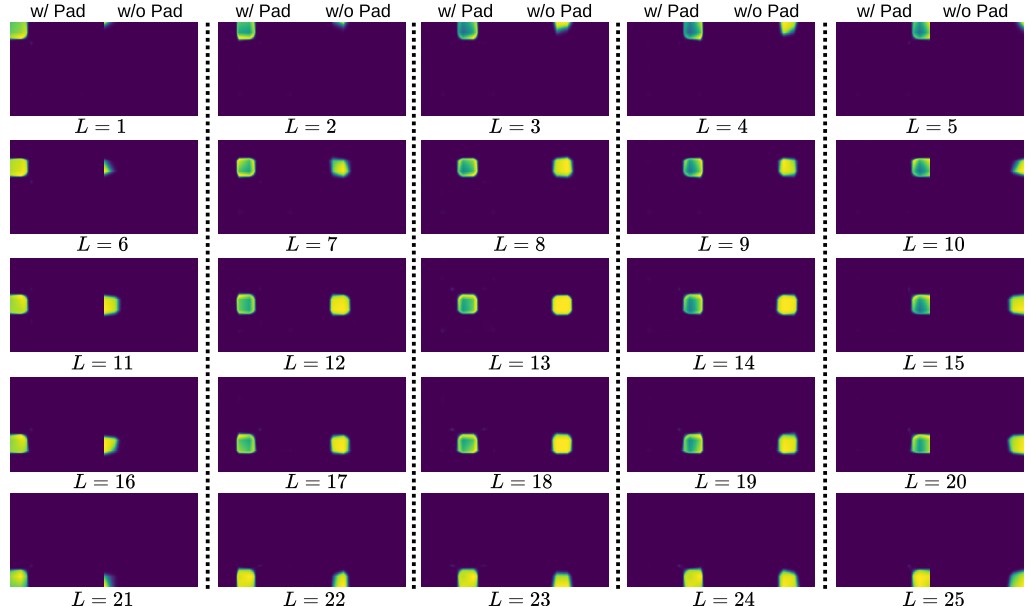

Figure 13: **Sample predictions of image segmentation** on all the locations of a $5 \times 5$ grid under the mean canvas setting. Confidence maps are plotted with the 'viridis' colormap, where yellow and dark blue indicates higher and lower confidence, respectively.

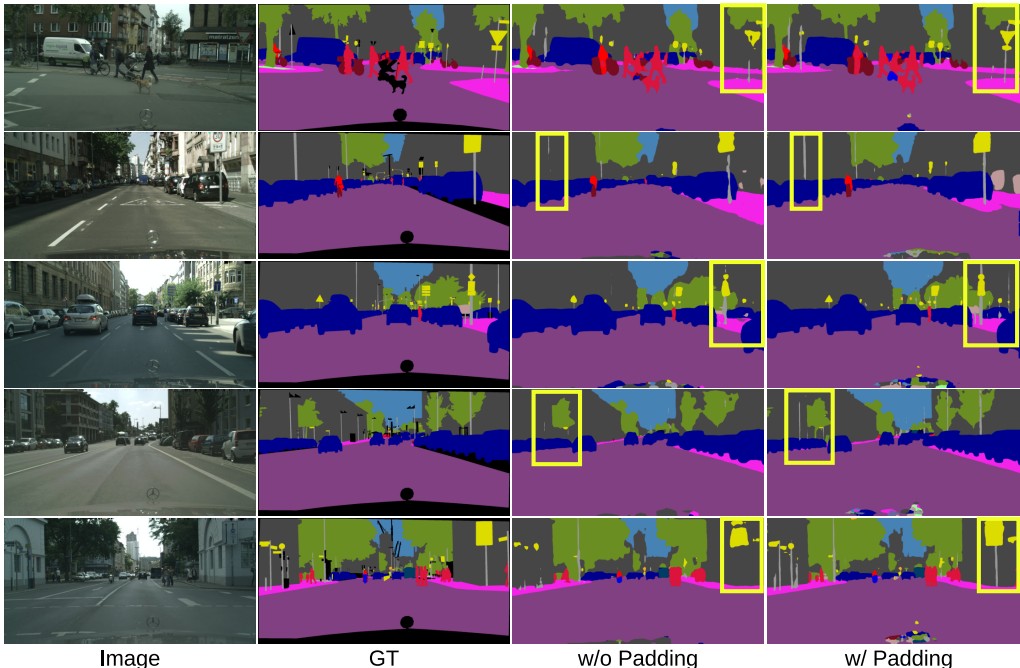

Figure 14: **Example predictions on the Cityscapes validation set** when training w/ and w/o *padding*. Best viewed with zoom.

Table 11: **IoU comparison of DeepLabv3** for semantic segmentation task with three different *padding* (Zero, Reflect, and No pad) settings.

| Eval. Region | Zero Pad | Reflect | No Pad |
|---|---|---|---|
| 0%- 5% | 72.6 | 71.9 | 63.7 |
| 5%- 10% | 72.7 | 72.0 | 66.4 |
| 10%- 15% | 73.8 | 73.7 | 67.2 |
| 15%- 20% | 73.9 | 74.1 | 67.9 |
| 20%- 25% | 74.7 | 74.8 | 68.5 |
| 25%- 30% | 75.3 | 75.4 | 69.6 |
| 30%- 35% | 75.1 | 75.2 | 69.4 |
| 35%- 40% | 74.7 | 75.2 | 69.3 |
| 40%- 45% | 74.4 | 74.8 | 69.2 |
| 45%- 50% | 74.2 | 74.5 | 69.4 |
| 50%- 55% | 74.4 | 74.9 | 69.8 |
| 55%- 60% | 74.3 | 74.8 | 69.7 |
| 60%- 65% | 73.8 | 74.3 | 69.2 |
| 65%- 70% | 73.8 | 74.4 | 68.8 |
| 70%- 75% | 73.9 | 74.5 | 68.9 |
| 75%- 80% | 73.8 | 74.4 | 69.2 |
| 80%- 85% | 73.5 | 74.1 | 68.1 |
| 85%- 90% | 71.4 | 71.9 | 65.1 |
| 90%- 95% | 71.3 | 72.0 | 64.2 |
| 95%- 100% | 69.7 | 70.1 | 70.2 |
| **Overall** | **74.0** | 73.9 | **69.1** |

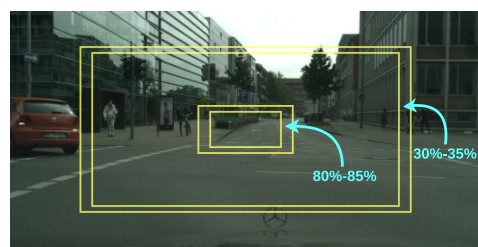

Figure 15: An illustration of the evaluation regions used for the analysis in Table 11 and Fig. 7.

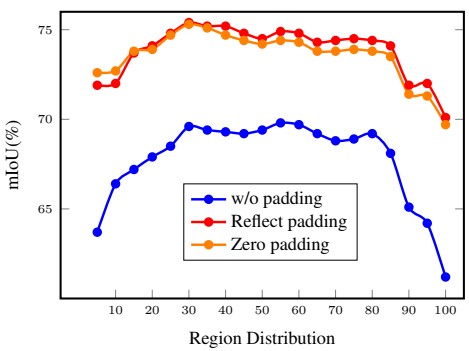

Figure 16: Performance comparison of DeepLabv3 network with respect to various image regions and padding settings.

## A.8 Extended Boundary Effect Analysis on Cityscapes Dataset

We continue to investigate the impact that zero padding has on the ability of a strong and deep CNN to segment objects near the image boundary. Results shown use the same network and training settings as in Sec. 6 of the main paper, on the Cityscapes (Cordts et al., 2016) dataset. We first show additional qualitative examples in Fig. 14, which clearly shows a large reduction in performance at locations near the border when no padding is used, particularly for thin objects (e.g., street lamps or column poles).

We present additional results (see Table 11 and Fig. 16) of the analysis presented in Sec. 6 (semantic segmentation) in the main paper. Fig. 15 shows sample evaluation regions used for this analysis. The no padding case has a steeper drop-off in performance as regions of evaluation get closer to the image boundary. Note how, in all cases, the performance increases from the border to the inner 25%, at which point the performance is somewhat stagnant until it reaches the innermost 80%.

Surprisingly, we also observe a *steeper* drop off in the middle of the image for the no padding case, supporting our hypothesis that boundary effects play a role at all regions of the image without the use of padding. We believe the drop in performance at the center regions is due to Cityscapes being an automotive-centric dataset, where pixels at the center of the image are often at large distances away from the camera, unless the vehicle collecting the data has an object directly in front of it.

## A.9 Canvas Analysis: Cutout & Adversarial Robustness

Figure 17 shows two training examples of Cutout strategy. Following Cutout, we simply place a rectangular mask (black and white) over a random region during the training. Note that we evaluate on the standard PASCAL VOC 2012 validation images.

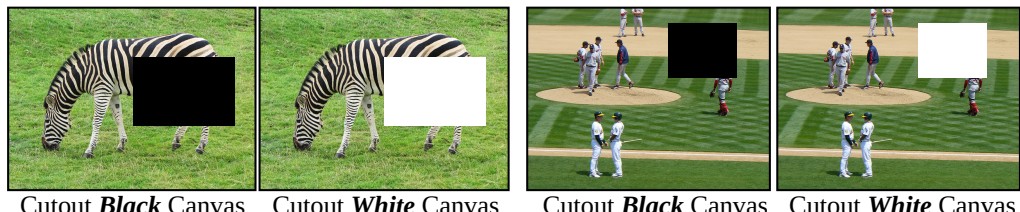

Cutout **Black** Canvas    Cutout **White** Canvas    Cutout **Black** Canvas    Cutout **White** Canvas

Figure 17: Sample training images generated using Cutout under two different canvases.

