# OpenReview forum: "Boundary Effects in CNNs: Feature or Bug?"
_ICLR.cc/2021/Conference — Reject_

### Official Review · AnonReviewer4 · 2020-10-23
**Questionable design and too counterintuitive results**

**Rating:** 3
**Confidence:** 4

**Review:**

The paper seeks to understand how different padding modes and canvas colors affect the performance of a convolutional neural network in classification and semantic segmentation tasks. The question seems somewhat strange - surely a network should be able to counteract a consistent change in padding or background color. If there was a strong effect it would be an interesting finding indeed. Unfortunately, the paper fails to convince that any but the most obvious effects exist.

There is an underlying assumption that knowledge of absolute spatial location should be helpful in some tasks and unhelpful in others. Why should absolute position matter for CIFAR-10 classification or semantic segmentation? Before jumping to this conclusion, this should be measured directly, e.g., by embedding horizontal and vertical positions as constant feature maps at every convolution.

The findings in Section 2 are not very surprising. Assuming that images are not mostly black, the greatest contrast is found against a constant black background, and partial convolution only enhances this effect - it is equivalent to zero padding but brightening the image artificially when the convolution kernel overlaps it only partially.

Table 2 shows that classification results vary wildly between padding and no padding, when comparing black canvas and zero padding. This makes no sense to me. Especially in large grids, a picture embedded somewhere else than the edge of a black canvas, without padding, should be fundamentally the same as having just the picture and zero padding. ResNet-18 that the paper uses has a global pooling layer at the end, ensuring that global position information is certain to be lost at that point at the latest.

I'm extremely concerned about the paper's decision to use bilinear interpolation to match the resolution with skip connection in a ResNet architecture, when padding is not used. My understanding is that the convolution branch output is upscaled to the skip branch resolution before composition. This introduces a scaling factor to the convolution branch, and therefore moves the contents of feature maps towards the edges. The composite will thus not have the features from the two branches line up with each other  anymore. This shifting effect is largest near the edges and smallest near the center, which matches with the observed performance characteristics. The subsequent convolution layers may be able to undo some of this shifting, but only at the cost of location-dependent kernels that are tailored to fit the offset  caused at different parts of the image. I'm inclined to believe that many of the alleged problems with non-padded convolutions are due to this design. For example, Table 10 suggests that the padding itself is a much more important factor than if padding is with zeros or a reflection, which makes little sense if locating the boundary were the critical aspect.

The paper is mostly clearly written, but some of the terminology is confusing. Section 3.1 talks about location-dependent classification, whereas the goal is to have classification be independent of location.  There are other cases of similar dependence vs independence confusion. What Horizontal and Gaussian mean in Table 1 should be explained. On page 7, formula $\frac{k-1}{k}$ should read $\frac{k^2-1}{k^2}$. In Figure 6, it is unclear how the activations are mapped to grayscale values. Regardless, high or low activations do not mean much with ReLU activation function, as any scaling can be counteracted in the weights of the subsequent convolution kernels, so the relative strengths of activations may not carry much information. In Figure 12, are the w/pad and w/o pad flipped? The effect seems inverse to Figure 1.

In summary, I find the findings questionable and too counterintuitive - e.g., the big difference in classification power between black background vs. zero padding when a small image is embedded in a large black canvas. The paper fails to demonstrate convincingly that these findings are real instead of artifacts of the experimental design.

Pros: mostly clearly written, many experiments.

Cons: dubious choices for network design and experiments, counterintuitive results that are not properly
explained or analyzed.

** UPDATE after reading other reviews, author responses, and revised paper **

I'm unconvinced by the arguments related to the resizing of feature maps between ResNet branches. The added experiment in A.1 does not address the question of misaligning the features between skip branch and convolution branch.

Figure 8 suggests that the alignment with input is well preserved with resizing (i.e., Fig. 8(a, d) look similar). The illustration is misleading because it uses a kernel with constant weights and *extremely* oversaturates the colors. In other circumstances, we would see that each corner in Fig. 8(c, d) is different. This is because Fig. 8(b) will contain four identical blobs after the convolution, and Fig. 8(c) is by definition a central crop of Fig. 8(b). Thus, the bits we see in Fig. 8(c) are different quadrants of this blob, and they are in fact all different. This leads to the misalignment between skip and convolution branches, so that subsequent processing will see different data at each of the four corners. The figure makes it look like there are only small differences by oversaturating the colors to the point that only the footprint of the convolution can be seen. Using a non-symmetrical convolution kernel would have also highlighted the differences.

The experiment also misses the larger point: The different architectures may cause differences in the results that exceed the effects of the padding itself. The paper claims to measure the latter, but I believe both AR2 and I are concerned that the paper may be measuring the architectural effects instead. As this was my main concern about the experimental design, my confidence in the paper's results is not increased, and my rating is not changed. If anything, I'd be inclined to lower my score because of how misleading the added Fig. 8 is.

---

> ### Author Response · Authors · 2020-11-20
> **Author Response to AR4**
>
>
> Dear Reviewer 4,
>
> Thank you for reviewing our paper and constructive feedback about our work. We agree with AR4 that revealing an effect between a change in padding or background color would be an interesting finding. One of the most salient examples of this finding is presented in Table 11 (previously Table 9), which shows a clear and significant difference in performance between a black and mean canvas. While we have not explained exactly why this difference exists, we have demonstrated this effect throughout our paper (e.g., Fig. 1, Table 2, Fig. 4 (left), Table 6) which is a valuable contribution that should be brought to light.
>
> - Main concerns
>
> AR4 takes two main issues with the paper: (i) “the paper fails to convince that any but the most obvious effects exist” and (ii) “too many of the results are counterintuitive”. We are unsure as to how both of these can be weaknesses of the paper as they seem to be in conflict with one another. We agree that some results found in our paper are quite counterintuitive which is precisely why they are interesting and not obvious. The fact that padding and canvases can cause significant performance differences in various tasks is an important and largely understudied phenomenon that has large implications for the community.
>
> - Absolute position hurts or helps different tasks
>
> A CNN's ability to leverage border effects could be considered a `bug' when a task requires translation-invariance, e.g., texture recognition (Table 5(a)); however, it could be useful if the task relies on position information, e.g., semantic segmentation (Table 4(a)). Note that (Liu, 2020) shows that appending the embedding of the spatial position within each convolution significantly affects performance on a number of tasks.
>
> - Findings in Section 2
>
> We agree with AR4 that the results in Sec. 2 are ‘not very surprising’. We use Sec. 2 (and the fact that zero padding delivers the most position information) to justify the use of zero-padding in the rest of the experimental sections (Secs. 3-5).
>
> - Table 2 classification results
>
> We have shown that both the padding and canvas color affect the CNN’s prediction separately from one another (see Fig. 1). This is why the results between the black canvas w/ and w/o zero padding differ since zero padding injects position information (Islam, 2020).
>
> - ResNet-18 with global pooling layer
>
> (Kayhan, 2020) (see Fig. 1) have shown that despite the pooling layer in a CNN, it can still learn to classify images based on their absolute position, contrasting the widely held assumption that the global pooling layer removes all position information.
>
> - ResNet No Padding Implementation
>
> There are many ways to implement a ResNet w/o padding. Since information is always lost at the border, we believe there is no 'perfectly fair’ way to compare padding and no padding ResNets. Note that the no padding comparisons are included for completeness to contrast the difference in the border effects between padding and no padding. However, we believe the spatial relationship is best retained using bilinear interpolation (BI). To this end, we conduct an experiment to validate the use of BI and have included a comparison between cropping (AR2’s suggestion) and BI (ours) in Appendix A.1 of the revised manuscript. Our conclusion is that BI provides a more fair comparison between padding and no padding ResNets as cropping discards more content near the border.
>
> - Horizontal and Gaussian Mean in Table 1
>
> These are the normalized gradient-like position map used as synthetic groundtruth to validate the existence of position information in (Islam, 2020). We have revised the manuscript with an explanation.
>
> - Location dependent vs. independent of location
>
> We agree with AR4 that the goal is to have classification be independent of location. We used ‘location dependent’ to highlight that the task is dependent on location (i.e., the network must be able to accumulate evidence of objects existing at every location) while the final prediction is independent.
>
> - Formula (k^2-1)/k^2
>
> Thank you for pointing this out. We have revised the manuscript accordingly.
>
> - Grayscale Mapping
>
> We obtain the activations from conv layer (not ReLU) and normalize the activations between 0-1 before plotting. Thus, we believe relative strengths of activations carry significant information about how filters are learned with different canvases. We have updated Sec. 5 accordingly in the revised manuscript.
>
> - Figure 12 clarification
>
> No, the w/ pad and w/o pad are not flipped in Fig. 12 (Fig. 13 in the revised manuscript). In both figures, removing the padding significantly reduces the confidence at the border and some of the predictions near the center have higher confidence than the network with padding.
>
> Thank you again for reviewing our paper. We would appreciate it if you could let us know if this changes your assessment.

---

> ### Author Response · Authors · 2020-11-23
> **Request for Discussion**
>
> Dear Reviewer 4,
>
> We believe that we’ve addressed all of your concerns in the response below and revised paper draft. Please let us know if there are any remaining concerns or questions! We would especially appreciate a response before the discussion period ends on Tuesday so that we can clarify any further questions.

---

### Official Review · AnonReviewer1 · 2020-10-28
**This paper provides valuable insight about padding in CNNs through extensive experiments and is well written.**

**Rating:** 7
**Confidence:** 4

**Review:**

This paper studies the effect of padding on the Convolutional Neural Network. The authors try to answer the following questions: 1) what type of padding provides the most position information, 2) does the background value affects model accuracy when processing a patch on a canvas, 3) which part of the image suffers the most from the boundary effect, and 4) whether the position information provided by padding improves or degrades model performance. In order to answer these questions, the authors design multiple tasks and perform extensive experiments. The empirical results show that: 1) zero-padding provides the most location information compared with other common padding methods, 2) the background value of the canvas do affects the accuracy when processing a patch, 3) the boundary effect is not specific to the image boundary---the model is affected by the boundary over the entire image, and 4) the effect of padding on model accuracy depends on the task.

This paper studies an empirically important yet under studied problem and is well written. The target questions / hypothesis are well motivated, and the experiment designs are reasonable and help to answer the target question. The extensive experiments provide valuable insight into the model properties and may benefit both fundamental CNN research and CNN applications in the future. Nevertheless, there are some problems that are not sufficiently addressed in the paper and may be improved.
First of all, an implicit hypothesis in the experiments is that the boundary effect is generic to the architecture. However, my conjecture is that how much padding affects the model strongly depends on the overall model architecture such as the readout architecture (e.g. avg pooling, max pooling, fc-layer), the receptive field of the neurons, etc. The correlation between model architecture and boundary effect is not discussed in the paper, and the implementation details are not clearly described to estimate how much it may affect the conclusion.

Second, as an empirical study with newly defined tasks, it is important to demonstrate whether the results are significant. More specifically, it would be helpful if the authors can 1) include intuitive baselines for reference performance (e.g. a model on (k+1)^2 patches instead of k^2), 2) show the learning curve / variance etc. to demonstrate whether the difference is meaningful, and 3) provide a more complete description about the experiment protocol (eg. how were the meta-parameters determined, whether the results are affected by the meta-parameters).

Third, some of the arguments are based on qualitative results. However, the figures are not clear enough and it is hard to follow the arguments. Also, qualitative results do not provide strong enough evidence for these arguments, and it would be more persuasive if the authors can provide quantitative analysis instead and use qualitative results as a support.
Finally, given that a learned representation can be distributed, I don't think the dimensionality estimation (Table 3) alone provides useful information. The argument implicitly assumes that 1) all other dimensions are independent of location information, 2) the location information cannot be reduced to a lower dimensional space. These assumptions are not verified.

The rebuttal provide valuable information that strengthens the original paper. While some of my concerns are not resolved in the rebuttal, they require additional experiments and may be beyond the scope of rebuttal.

---

> ### Author Response · Authors · 2020-11-20
> **Author Response to AR1**
>
>
> Dear Reviewer 1,
>
> Thank you very much for your valuable feedback. We appreciate your assessment of our work as  ‘an empirically important yet under studied problem’, ‘hypothesis are well motivated’, and ‘extensive experiments provide valuable insight into the model properties and may benefit both fundamental CNN research and CNN applications in the future’.
>
> - Correlation between network architecture and boundary effects
>
> We fully agree with AR1 that there is a correlation between network architecture and boundary effects and thank AR1 for the suggestion to discuss this further. While we use three different baseline architectures to validate the boundary effects throughout the paper (i.e., ResNet and VGG-5 in Secs. 2-6, DeepLabv3 in Sec. 6 are all widely used architectures with common layer types), we agree that a more detailed analysis of padding with respect to specific architectures is important and we will explore this in future research.
>
> - Intuitive baselines for reference performance
>
> Note: We apologize for our uncertainty, but unfortunately this question is slightly ambiguous to us. Here is how we interpret it two different ways: (i) Training a model using k^2 patches and validating it using (k+1)^2 patches. Or (ii) Training and testing a model using (k+1)^2 patches where k={3,5,7…} We welcome clarification if we have misinterpreted it.
>
> Answer: We conduct two sets of experiments to generate the suggested reference performance and include the results here:
>
> (i) Train a segmentation network with 5x5 and 7x7 patches and validate with 6x6 and 8x8 patches respectively. Results are:
>        Train on 5x5 Grid and evaluated on 6x6 Grid with black canvas + padding : 65.7
>        Train on 5x5 Grid and evaluated on 6x6 Grid with black canvas + w/o padding : 51.9
>
>       Train on 7x7 Grid and evaluated on 8x8 Grid with black canvas + padding : 66
>       Train on 7x7 Grid and evaluated on 8x8 Grid with black canvas + w/o padding : 50.9
>
> We can see from these results that the networks trained and evaluated on different grid sizes can perform the task well, and only suffer from minor drops in performance.
>
> (ii) Train a segmentation network with 4^2 and 6^2 patches separately and validated on 4^2 and 6^2 patches respectively: Results:
>               4x4 Grid: w/ padding Black canvas: 68.8
>               4x4 Grid: w/o padding Black canvas: 58.1
>               6x6 Grid: w/ padding Black canvas: 68.1
>               6x6 Grid: w/o padding Black canvas: 54.9
>
> We have included these reference results in Table 9 of the Appendix in the revised manuscript.
>
> In addition, we agree with AR1 that adding the variance would give more insight to the meaningfulness of the results, and will include this for all results in Table 2 to the final manuscript. However, we run a few experiments to provide some initial insight.
>
> 5x5 Black canvas w/ padding results of 5 different runs:  68.5, 67.5, 68.6, 67.7, 67.2
> Variance: 0.385
>
> 5x5 Black canvas w/o padding results of 5 different runs: 53.5, 54.92, 55.5, 54.4, 55.1
> Variance: 0.592
>
> 5x5 white canvas w/ padding results of 5 different runs:  64.2, 65.5, 64.6, 64.9, 64.3
> Variance: 0.275
>
> In conclusion, we can see that the variance is less than 1% for these experiments which we believe increases the confidence of the results. We thank AR1 for the suggestion to provide a more detailed description about the experiment protocol and include more robust baseline results.
>
> - Qualitative results do not provide strong enough evidence for these arguments
>
> We agree with AR1 that qualitative results alone are not sufficient evidence to validate any finding with absolute certainty, however, we aimed to provide quantitative results where necessary in our experiments. Note that additional quantitative analysis can be found in the Appendix, such as evaluating the per-class border effect for a segmentation network in Table 11 (previously Table 9) on the CIFAR-10 dataset and another region based analysis on the cityscapes dataset in Table 12 (previously Table 10).
>
> - Distributed learned representation vs. dimensionality estimation (Table 3)
>
> We do not need an estimate of the absolute MI for our purpose. Only the differences between the mutual information for position and semantic class for image pairs are used to quantify the ratio of location and semantic-specific neurons. Therefore, the relative difference is still meaningful and only the absolute numbers might not be. We agree that this should be made more clear, and have updated A.5 (previously A.4) in the revised manuscript.
>
> Again, thank you very much for reviewing our paper and for the constructive feedback! We would appreciate it if you could let us know whether our response clarified your concerns, or if there are any other questions you have.

---

> ### Author Response · Authors · 2020-11-23
> **Request for Discussion**
>
> Dear Reviewer 1,
>
> We believe that we’ve addressed all of your concerns in the response below and revised paper draft. Please let us know if there are any remaining concerns or questions! We would especially appreciate a response before the discussion period ends on Tuesday so that we can clarify any further questions.

---

### Official Review · AnonReviewer3 · 2020-10-28
**Interesting paper on boarder padding analysis**

**Rating:** 8
**Confidence:** 3

**Review:**

The topic of the paper is interesting and important. The padding strategy seems to be a small but largely overlooked aspect in CNN learning. While many papers attempt to reduce/improve the effect of position in images. This paper gives a different perspective on the padding patterns that essentially causes these position sensitivity.

The paper is easy to follow: the hypothesis at the beginning are important and interesting. And the hypothesis are quite clearly verified with experiments.

The Hypothesis raised in this paper are important factors for training networks for various tasks. especially for H5 and H2. And the paper gives useful conclusions based on experiments.   H4 is a bit unnecessary as it is common knowledge that as layer goes deeper, it has a larger receptive field thus even boarder information will be included in the center regions.

The current analysis are based on CIFAR, it would be more convincing if these hypothesis are further validated on larger datasets such as imagenet or coco (for segmentation).

A minor suggestion, it would be more informative if the author in related work/analysis compare with the positioning methods used in transformers.

---

> ### Author Response · Authors · 2020-11-20
> **Author Response to AR3**
>
> Dear Reviewer 3,
>
> Thank you very much for reviewing our paper and your positive comments about our work “The topic of the paper is interesting and important, “This paper gives a different perspective on the padding patterns that essentially causes these position sensitivity”, “The paper is easy to follow: the hypothesis at the beginning are important and interesting. And the hypotheses are quite clearly verified with experiments.”, “the paper gives useful conclusions based on experiments”. We respond to your comments individually below.
>
> - Experiments on larger datasets
>
> We conduct the analysis based on CIFAR in order to control various confounding variables and better test the underlying hypotheses. However, we validate the findings of each hypothesis on real world datasets, namely, CityScapes, GTOS-M, DTD, and PASCAL VOC 2012. (Sec. 6 in the paper). Our conclusion from the experiments conducted on these datasets is that position information has an effect on all tasks, but can be harmful or beneficial to have it depending on the task.
>
> - Compare with the positioning methods used in transformers.
>
> We thank AR3 for the suggestion to include analysis of how positioning methods are used in transformers. Zero padding implicitly encodes position information within CNNs, however adding explicit positional encodings can improve performance in both vision and natural language processing tasks. For example, previous work (Liu, 2018b) proposed an effective explicit positional encoding (i.e., CoordConv, which appends the x and y coordinate onto the latent representation) to the convolutional layer and have shown that this additional position information improves the performance of CNNs on a number of tasks, including generative modelling, reinforcement learning, and object detection. We believe that the positional encoding found in transformers is similar to CoordConv in that the position of the input feature is explicitly given to the network as an additional feature in some manner. Note that there are a number of ways to encode this information which differ between natural language processing tasks and vision tasks. For example, the traditional way of generating the position encoding vector in transformers is by describing the position using a number of sin and cosine functions with frequencies corresponding to the input's position (Vaswani, 2017).
>
> Again, thank you very much for reviewing our paper and for the suggestions!
>
> (Vaswani, 2017) Vaswani et al. “Attention Is All You Need”. NeurIPS 2017.

---

> ### Author Response · Authors · 2020-11-23
> **Request for Discussion**
>
> Dear Reviewer 3,
>
> We believe that we’ve addressed all of your concerns in the response below and revised paper draft. Please let us know if there are any remaining concerns or questions! We would especially appreciate a response before the discussion period ends on Tuesday so that we can clarify any further questions.

---

### Official Review · AnonReviewer2 · 2020-10-28
**This manuscript investigates the important topic of padding in CNNs. The review points out some potential issues in the experimental evaluation.**

**Rating:** 3
**Confidence:** 4

**Review:**

Summary of the paper:

This submission studies the effect of zero/one padding in convolutional networks when using images that are pasted on a canvas. The main thesis is that zero padding induces absolute position information and that this leads to better performance in many cases. The study is based on segmentation and classification where the input image is placed on a background canvas.

Update after Rebuttal:

I do appreciate the argument given in appendix A.1 about the relative position that changes with cropping and that boundary information may get lost. However, it is not clear to me if that change in relative position matters because the network could take this into account. However, the issue that the feature maps between the shortcut and residual connection do not align with the bilinear interpolation seems to be much harder take into account for the network. To me it seems an actual example on the ImageNet classification (i.e. Table 2) that shows that the degradation in performance is not due to the resampling/misalignment of the feature maps in the ResNet would be very important. Moreover, a related problem is that it seems plausible that a network could extract position information from the spatially varying misalignment of the feature maps (in the image center there is no misalignment and on the border there is 1px (for 3x3 conv). The amount would reveal the position). Therefore an experiment that shows that this does not happen in practice would also be important. This is the same major concern I share with AR4 even after the rebuttal.


Strengths:

Padding is omi-present in current convolutional and "fully" convolutional architectures but its effect is not studied well.
The manuscript clearly states the hypothesis that are investigated.

Weaknesses:

The motivation for using canvases in the way done within this manuscript is unclear to me. The introduction states this is due to the fact that images need to be rectangular to be processed with a CNN. While in practice mostly rectangular images are used with CNNs it seems to me that convolutions could also be applied to non-rectangular images if required (with custom implementation). However, more importantly the manuscript does not just pad images to be rectangular but instead pastes a much smaller image onto a large canvas. This is not clearly motivated in the current form of the manuscript as far as I can tell.

In Section 3 it is unclear if one network is trained for all grid resolutions or if the networks are trained per grid resolution.

Unclear how no padding is conducted in ResNet. Section 3 indicates that a standard ResNet-18 is modified to a no padding version by removing the padding from the convolutions and using bilinear resize to match the sizes. The way I understand this is that for example an NxN image is convolved without padding and hence the output will be an (N-k) x (N-k) image, now either the output of the convolution or the original NxN image is resized to match the other resolution before adding the two. This seems like a very problematic implementation as in either version the spatial alignment between the two is not preserved and importantly the misalignment is spatially varying while the convolution kernel will not vary spatially. In my mind a no padding implementation of a ResNet would simply crop the original image in the consistent way to how the convolution does not produce outputs for the boundary regions. If my understanding of this implementation is correct I feel that this is a major flaw in this submission.

Do the no padding segmentation networks contain upsampling operations and or u-net type skip connections? How are the no padding versions of these operations implemented? Especially for the city scapes segmentation results it is unclear to me how a full image segmentation was obtained without using any form of padding or related mechanism of retaining the information around the image boundary.

In Table 3 the white padding seems to contain stronger location information than the black padding which contradicts the main thesis of the submission that says black padding contains more location information. However, Table 3 uses different grid sizes for black and white padding. Are the different grid sizes the reason for this discrepancy? What is the motivation for using different grid sizes?

One point I find missing in this submission is that no padding networks will have spatially smaller feature maps which means less information might be stored in the intermediate layers. This could be a reason why the no-padding versions of Table 4 underperform compared to reflection padding.

What happens for networks where the input/output resolution varies, i.e. "fully" convolutional networks? In these cases absolute position information intuitively would prevent generalization to different image sizes.

Reason for score:

This submission discusses an important and relevant topic. However, at this point I believe that there are potentially flaws in the experiments which might significantly influence the findings. Therefore the current rejection rating. If the rebuttal could further discuss the points in the weaknesses section and explain in detail how the no-padding versions were implemented that would be helpful to further assess the correctness of the above assessment.

---

> ### Author Response · Authors · 2020-11-20
> **Author Response to AR2**
>
>
> Dear Reviewer 2,
>
> Thank you for reviewing our paper. We appreciate your suggestions and positive comments “clearly states the hypothesis that are investigated”. We provide a point-by-point response to each concern below.
>
> - Motivation for rectangular images
>
> While we agree with you that there could be ways to apply CNNs on irregular non-rectangular images, we are interested in studying the most standard setting in computer vision, convolutions on a rigid image format. Thus, we believe that rectangular input is a more important topic to study for the community as it is far more widely used.
>
> - Pasting image onto canvas
>
> Previous work (Kayhan, 2020) conducted an experiment by pasting an image patch on a black canvas to determine if a CNN can classify the image location for different resolutions (i.e., top left or bottom right). (Islam, 2020) showed that zero padding (i.e., black) significantly increases the amount of position information encoded in the network. This suggests the border color may be playing a role in the CNNs position encoding. Thus, we paste patches on various canvas colors and sizes with the motivation of evaluating whether the canvas color had an effect on the amount of position information encoded at various distances to the boundary.
>
> - Clarification on networks trained per grid resolution
>
> The networks are trained per-grid resolution (i.e., for grid sizes 3x3, 5x5, two separate ResNet-18’s are trained).
>
> - No padding clarification
>
> AR2 raised a major concern regarding the no-padding baseline implementation. You are correct in your understanding of our implementation. Note that the no padding comparisons are included for completeness to contrast the difference in the border effects between padding and no padding. There are a variety of ways to implement a ResNet w/o padding however we apply interpolation because the spatial relationship will be best retained compared to other methods. AR2 suggests an alternative way to implement a ResNet with no padding is to crop the input to the same size as the output. We appreciate the suggestion and, to this end, conduct an experiment to validate the use of our implementation and include a comparison between cropping (AR2’s suggestion) and bilinear interpolation (ours) in Appendix A.1 of the revised manuscript. We generate a toy image with four 10x10 colored squares, each at one corner of a 100x100 black image, and pass this through a conv layer with a 21x21 kernel. We then compare the methods of (i) cropping the input to match the output resolution (80x80) and (ii) bilinear interpolation of the output. We conclude that cropping discards more of the content near the border and that bilinear interpolation also keeps the spatial alignment more consistent. Therefore we believe our use of bilinear interpolation provides the most fair implementation when comparing padding and no padding ResNets.
>
> - No padding segmentation networks
>
> Yes, the no-padding segmentation networks contain upsampling. The no padding DeepLabv3-ResNet50 (DLR50) network trained on CityScapes used the same bilinear interpolation strategy. We follow existing work (e.g., Chen, 2018) and upsample the prediction to the target resolution using bilinear interpolation. The no padding ResNet suffers more from boundary effects which is why DLR50 trained w/o padding fails to generate accurate predictions, particularly near the border (see Figs. 7 and 14 (previously 13)).
>
> - Table 3 white vs black canvas differences and motivation for different grid sizes
>
> Yes, this is due to the smaller grid-size used for the white canvas in Table 3. Table 10 in the revised manuscript (previously Table 8) shows a 7x7 grid w/ white canvas and shows that the white canvas supplies less positional information. We agree that this may be confusing to the reader, and have changed Table 3 accordingly in the revised manuscript.
>
> - No padding will have smaller feature maps
>
> We do not decrease the feature map size for fair comparison, and instead use bilinear interpolation. You are correct in your assessment that less information may be stored in the intermediate features. As discussed in the original manuscript (i.e., first paragraph of Sec. 1, last sentence of Sec. 6, Texture Recognition), CNNs suffer from border effects without padding since the kernel’s support does not cover the entire image domain.
>
> - Does position information generalize to different image sizes?
>
> A significant change in image scale may prevent the generalizability of neural networks and impact the performance of a network on different tasks. This depends on the effective receptive field of the network relative to the input size, which is explored in (Kayhan, 2020) (Fig. 4).
>
> Again, thank you for reviewing our paper and for your valuable suggestions! We would appreciate it if you could let us know whether our response clarified your concerns and if this changes your assessment of our work, or if there are any other questions you have.

---

> ### Author Response · Authors · 2020-11-23
> **Request for Discussion**
>
> Dear Reviewer 2,
>
> We believe that we’ve addressed all of your concerns in the response below and revised paper draft. Please let us know if there are any remaining concerns or questions! We would especially appreciate a response before the discussion period ends on Tuesday so that we can clarify any further questions.

---

### Decision · Program_Chairs · 2021-01-07
**Final Decision**

**Decision:**

Reject

**Comment:**

This paper explores the effects of padding in convnets used for various visual recognition tasks (classification, segmentation). This is an important and relevant design choice that is often overlooked, as noted by reviewers. However, I share the concerns of AR2 & AR4 with the evaluation. The design of the ResNet variant used for the "No Pad" baseline seems potentially fatally flawed: the bilinear upsampling used to match the feature map sizes for the residual addition results in a misalignment of the inputs with the outputs, which potentially explains the performance degradations seen throughout most experiments, as opposed to (or perhaps in addition to) the lack of positional information and border effects resulting from the NoPad scheme that is claimed as the reason for the performance drop. It is true that how to do this is an open question, as the authors argue in their added Appendix A.1, but I nonetheless share the reviewers' skepticism that the chosen approach will result in a meaningful comparison of the effect of padding and border effects. In fact, the results in Table 5 on texture recognition seem to suggest that "No Pad" approach may indeed be flawed, given that "No Pad" performs the worst, while "Reflect" padding performs best, even though both methods share the property that the network should have difficulty inferring positional information. Given the reliance on this dubious baseline throughout the results, I can't recommend the submission for acceptance in its current form. However, I still appreciate the direction of this work and hope the authors will consider resubmitting it after revising it in order to make the evaluation more convincing based on the reviewers' feedback.